# Incentivizing DINOv3 Adaptation for Medical Vision Tasks via Feature Disentanglement

**Zhicheng He**[1,*]  ID                                    E1583419@U.NUS.EDU
**Yibing Fu**[1,*]                                          YIBINGFU@U.NUS.EDU
**Yueming Jin**[1,†]                                         YMJIN@NUS.EDU.SG
[1] *National University of Singapore*

**Editors:** Accepted for publication at MIDL 2026

## Abstract

The emerging general vision foundation models such as DINOv3 have demonstrated remarkable representation learning capability in natural image domains. However, transferring these representations to medical imaging is challenging due to substantial domain discrepancies. To bridge this gap, parameter-efficient fine-tuning (PEFT) has emerged as a promising strategy to adapt these vision foundation models to medical vision tasks by updating only a small subset of parameters while preserving pretrained knowledge. Despite the efficiency, existing PEFT strategies overlook that pretrained features inherently interleave task-relevant semantics with task-irrelevant patterns and noise, potentially limiting effective adaptation in medical scenarios. To address this challenge, we propose DINOv3-FD, a task-oriented feature disentanglement framework that adapts DINOv3 to medical vision tasks. DINOv3-FD introduces a dual-stream adapter that separates features into task-relevant and task-irrelevant subspaces, reinforced by an orthogonality loss to encourage their mutual independence. Additionally, a distributional regularization loss drives the task-irrelevant branch toward task-agnostic predictions, discouraging it from encoding task-specific semantics. Consequently, the task-relevant stream is encouraged to retain more discriminative representations that facilitate downstream medical tasks. Experimental results show that DINOv3-FD outperforms other PEFT strategies over three medical classification tasks, demonstrating the effectiveness of feature disentanglement. Our code is available at https://github.com/hezhicheng2002/DINOv3-FD.

**Keywords:** Feature Disentanglement, Representation Learning, Medical Image Classification

## 1. Introduction

Recent advancements of general vision foundation models (GVFMs) which are pretrained on web-scale natural images have been revolutionizing the computer vision domain. Models such as MAE (He et al., 2022), DINO (Caron et al., 2021), and iBOT (Zhou et al., 2022) provide strong visual priors for downstream tasks, significantly enhancing their performance across a broad spectrum of visual tasks, including classification, segmentation, detection, and so on. More recently, DINOv3 (Siméoni et al., 2025) has further unleashed the potential of self-distillation strategies by tailoring the pretraining strategy over web-scale images,

---

* Contributed equally
† Corresponding author

demonstrating remarkable performance and scalability. Although GVFMs provide powerful representations in natural image domains, their performances commonly diminish on medical imaging tasks because of significant domain shifts.

To narrow this gap, the medical AI community has increasingly focused on adapting GVFMs pretrained on natural images to medical vision tasks through parameter-efficient fine-tuning (PEFT) strategies. Rather than updating all model parameters, PEFT strategies such as Low-Rank Adaptation (LoRA) (Hu et al., 2021) and Adapters (Houlsby et al., 2019) selectively optimize only a small subset of parameters while keeping the majority frozen. These approaches effectively preserve the general visual priors encoded in the pretrained models while enabling efficient adaptation to medical downstream tasks. Moreover, it substantially reduces computational overhead and mitigates overfitting under low-data regimes, which is common in the medical imaging domain (Fu et al., 2025). These advantages make PEFT strategy a highly appealing paradigm, which intuitively raises a question- ***How can the latest DINOv3 be effectively adapted to medical vision tasks?***

Previous researches have demonstrated the effectiveness of adapting GVFMs to medical imaging tasks via PEFT strategies. For instance, Veasey and Amini leverage LoRA to adapt GVFMs for lung nodule malignancy classification, achieving improved performance with substantially fewer trainable parameters (Veasey and Amini, 2025). Wu *et al.* proposes Medical SAM Adapter, which incorporates lightweight adapters to adapt SAM for 2D and 3D medical image segmentation tasks (Wu et al., 2025). Despite the efficiency, such existing PEFT methods do not explicitly consider how the features extracted by GVFMs contribute to the medical tasks, and instead apply uniform updates across the entire embedding or projection layers. However, features from GVFMs may contain a mixture of task-relevant, task-irrelevant, and even noisy information. Therefore, how to disentangle these components during adaptation in order to strengthen task-relevant features while suppressing irrelevant counterparts remains an important yet unexplored problem.

In this work, we propose DINOv3-FD, which adapts DINOv3 to medical vision tasks from the perspective of task-oriented feature disentanglement. Specifically, we develop a **dual-stream adapter framework** that decomposes the feature representations into task-relevant and task-irrelevant subspaces. To explicitly enforce this separation, we incorporate an **orthogonality loss** that promotes mutual independence between the two branches. Additionally, we innovate a **distributional regularization loss** on the task-irrelevant adapter, encouraging its predictions to approach a label-agnostic random distribution. This incentivizes the adapter to encode task-irrelevant features rather than task-discriminative semantics. By jointly leveraging these three components, DINOv3-FD enables DINOv3 to better preserve task-relevant representations while effectively isolating noisy or task-irrelevant features. We comprehensively evaluate our method over three medical image classification tasks, demonstrating the superior performance of our proposed method. In conclusion, our contributions can be summarized as threefold:

- We propose DINOv3-FD, which introduces a dual-stream adapter to adapt DINOv3 to medical vision tasks in a task-oriented feature disentanglement manner.

- We propose an orthogonality loss across the two subspaces, which promotes mutual independence between the feature representations.

- We innovate a distributional regularization loss to the task-irrelevant branch, pushing its predictions toward a label-agnostic distribution to filter task-irrelevant features.

## 2. Related Works

### 2.1. Parameter-efficient fine-tuning

Parameter-efficient fine-tuning (PEFT) has emerged as an appealing strategy for adapting GVFM to data-constrained and heterogeneous medical imaging scenarios. Instead of updating all model parameters, PEFT selectively introduces lightweight modules or low-rank projection layers, achieving strong adaptation performance while retaining most pretrained prior knowledge. Early approaches began with adapter modules (Houlsby et al., 2019), which integrated lightweight adaptation layers to enable task-specific adaptations. Subsequently, LoRA (Hu et al., 2021) introduced low-rank updates to attention projections, which emerged as a milestone for fine-tuning foundation models. Following its success, numerous variants have been proposed to enhance LoRA's performance. For example, IA$^3$ (Liu et al., 2022) reduced trainable parameters through learned multiplicative vectors applied to key components of transformer blocks. LyCORIS (Yeh et al., 2024) expanded LoRA through hybrid pathways that make modulation more flexible. Building on this direction, VeRA (Kopiczko et al., 2024) reparameterized adaptation directly through optimizer states for further compression. PaCA (Woo et al., 2025) introduced parallel low-rank branches to refine attention and feed-forward transformations. However, these approaches mainly focus on how to design the adapter or projection layers while neglecting how to highlight task-relevant representations.

### 2.2. Feature Disentanglement

Feature disentanglement aims to reorganize latent feature representations such that task-specific information is distilled into a dedicated subspace, whereas task-irrelevant features or noises are isolated to another subspace since they may compromise the overall performance. This motivation has led to substantial advances in techniques aimed at reducing statistical dependencies within learned feature representations. Cogswell *et al.* introduced decorrelation penalties, exemplified by DeCov (Cogswell et al., 2015), which encouraged separation by directly suppressing correlated activations across feature dimensions. Building on this principle, kernel-based dependence measures such as HSIC (Gretton et al., 2005; Ma et al., 2020) expanded the concept of independence by capturing nonlinear interactions through kernel embeddings. Variational approaches like MINE (Belghazi et al., 2018) further generalized these ideas by estimating mutual information, supporting the modeling of richer and more diverse statistical dependencies. Building on these principles, recent studies developed architectures that explicitly divide representational roles across different model components. Invariant-learning frameworks such as IRM (Arjovsky et al., 2019) and IIV (Ahuja et al., 2021) promoted consistency across environments, pushing environment-dependent variation into distinct representational directions. In medical imaging, related ideas have been adapted to domain-specific sources of variation. For example, Wang *et al.* disentangled disease-related features from obscuring tissues via explicit factor separation (Wang et al., 2022). MIMM-X (Fay et al., 2025) reduced mutual dependence between causal and auxiliary

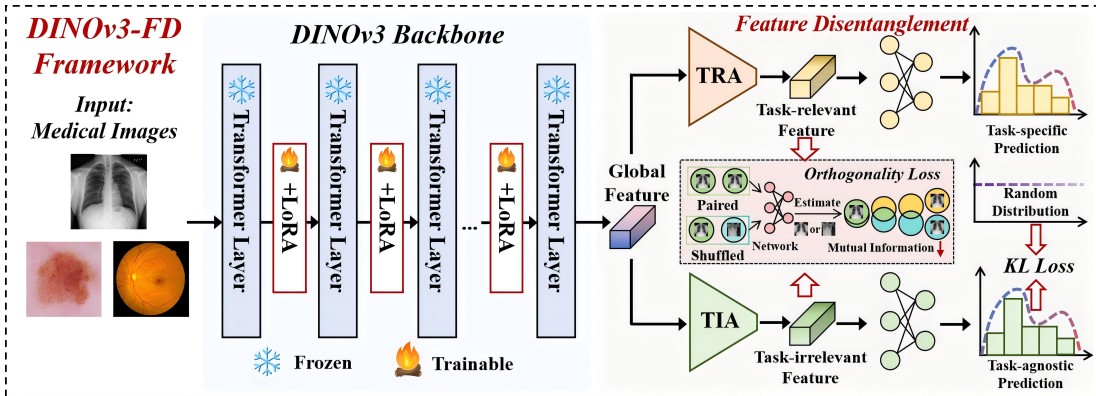

Figure 1: Overview of the DINOv3-FD framework. We leverage the DINOv3 integrated with LoRA layers as the vision encoder. The obtained global representation is routed to the task-relevant and irrelevant adapters for feature disentanglement. An orthogonality loss and a distributional regularization loss jointly boost feature disentanglement by promoting information independence and pushing the irrelevant branch toward label-agnostic predictions.

components. Recent segmentation frameworks further disentangle lesion-relevant features from background to reduce annotation noise (Xiong et al., 2024).

## 3. Method

### 3.1. Framework

As illustrated in Fig. 1, our DINOv3-FD framework adopts a frozen DINOv3 (Siméoni et al., 2025) model as the vision encoder, while incorporating lightweight LoRA (Hu et al., 2021) layers to facilitate efficient feature adaptation. Given an input image, the DINOv3 backbone yields a global representation, which serves as the input to our feature disentanglement module. Specifically, this representation is routed into two parallel adapters, i.e., a task-relevant adapter (TRA) and a task-irrelevant adapter (TIA). During training, we introduce an orthogonality loss and a distributional regularization loss to disentangle and strengthen task-relevant features within the TRA, while isolating task-irrelevant features via the TIA. During inference, only the TRA is retained for task-oriented adaptation.

### 3.2. Dual-stream Adapter Framework

In our DINOv3-FD framework, we leverage the latest vision foundation model DINOv3 and incorporate LoRA layers to build the vision encoder. Given a medical image, we extract the global representation $\boldsymbol{f}_{\text{cls}} \in \mathbb{R}^d$ from the encoder, where $d$ denotes the embedding dimension. Subsequently, we route $\boldsymbol{f}_{\text{cls}}$ to TRA and TIA to generate the task-relevant feature $\boldsymbol{f}_{\text{T}} \in \mathbb{R}^d$ and task-irrelevant feature $\boldsymbol{f}_{\text{I}} \in \mathbb{R}^d$. It is worth noting that both the TRA and TIA comprise two linear layers coupled with nonlinear activation functions, resulting in only minimal additional parameters overhead.

Subsequently, the task-relevant feature $\boldsymbol{f}_\mathrm{T}$ and task-irrelevant feature $\boldsymbol{f}_\mathrm{I}$ are processed by two linear classification heads to get the class decision scores as follows:

$$\hat{\boldsymbol{y}}_\mathrm{T} = g_\phi(\boldsymbol{f}_\mathrm{T}), \qquad \hat{\boldsymbol{y}}_\mathrm{I} = g_\psi(\boldsymbol{f}_\mathrm{I}), \tag{1}$$

where $g_\phi$ and $g_\psi$ denote the task-relevant and task-irrelevant prediction heads. $\hat{\boldsymbol{y}}_\mathrm{T}$ and $\hat{\boldsymbol{y}}_\mathrm{I}$ both reside in a $\mathbb{R}^c$ space, where $c$ denotes the target classes. This dual-stream adapter framework empowers the model to explicitly separate interleaved features into task-relevant and irrelevant subspaces.

### 3.3. Feature Disentanglement

To further promote mutual independence between $\boldsymbol{f}_\mathrm{T}$ and $\boldsymbol{f}_\mathrm{I}$, we introduce an orthogonality loss for feature disentanglement. Specifically, inspired by (Belghazi et al., 2018), we firstly introduce a statistical dependency estimator $E_\omega$, which is implemented as a small multilayer perceptron that takes the concatenated feature $[\boldsymbol{f}_T, \boldsymbol{f}_I]$ as input. Following the Donsker–Varadhan lower bound on mutual information (Belghazi et al., 2018), the dependence between two feature pathways is estimated as:

$$\mathcal{L}_\mathrm{ortho} = \mathbb{E}_{(\boldsymbol{f}_\mathrm{T}, \boldsymbol{f}_\mathrm{I})_\mathrm{pair} \sim \mathcal{D}_{T,I}}[E_\omega([\boldsymbol{f}_T, \boldsymbol{f}_I])] - \log \mathbb{E}_{(\boldsymbol{f}_\mathrm{T}, \boldsymbol{f}_\mathrm{I})_\mathrm{shuffle} \sim \mathcal{D}_T \mathcal{D}_I}[\exp(E_\omega([\boldsymbol{f}_T, \boldsymbol{f}_I]))]. \tag{2}$$

In the above equation, $(\boldsymbol{f}_\mathrm{T}, \boldsymbol{f}_\mathrm{I})_\mathrm{pair}$ refers to paired features, which originates from combining $\boldsymbol{f}_\mathrm{T}$ and $\boldsymbol{f}_\mathrm{I}$ from the same input image. Conversely, $(\boldsymbol{f}_\mathrm{T}, \boldsymbol{f}_\mathrm{I})_\mathrm{shuffle}$ denotes the shuffled features, which are formed by keeping $\boldsymbol{f}_\mathrm{T}$ fixed while randomly permuting $\boldsymbol{f}_\mathrm{I}$ from different images within a minibatch.

Mathematically, $(\boldsymbol{f}_\mathrm{T}, \boldsymbol{f}_\mathrm{I})_\mathrm{pair}$ approximates samples from the joint distribution $\mathcal{D}_{T,I}(\boldsymbol{f}_\mathrm{T}, \boldsymbol{f}_\mathrm{I})$ of the two subspaces. $(\boldsymbol{f}_\mathrm{T}, \boldsymbol{f}_\mathrm{I})_\mathrm{shuffle}$ approximates samples from the product of their marginals $\mathcal{D}_T(\boldsymbol{f}_\mathrm{T}) \mathcal{D}_I(\boldsymbol{f}_\mathrm{I})$. By minimizing $\mathcal{L}_\mathrm{ortho}$, the predicted scores of the paired tuples and shuffled tuples become increasingly similar, which implies that the joint distribution $\mathcal{D}_{T,I}(\boldsymbol{f}_\mathrm{T}, \boldsymbol{f}_\mathrm{I})$ becomes progressively harder to be distinguished from the product distribution $\mathcal{D}_T(\boldsymbol{f}_\mathrm{T}) \mathcal{D}_I(\boldsymbol{f}_\mathrm{I})$, which means low mutual information between $\mathcal{D}_T(\boldsymbol{f}_\mathrm{T})$ and $\mathcal{D}_I(\boldsymbol{f}_\mathrm{I})$. As a result, this optimization procedure promotes statistically independent and disentangled feature pathways.

### 3.4. Distribution Regularization

Furthermore, to ensure that the TIA subspace captures task-irrelevant representations, we introduce a distributional regularization loss. Specifically, the output $\hat{\boldsymbol{y}}_\mathrm{I}$ is encouraged to approach a uniform distribution $\mathcal{U}$ over the $c$ classes. This is achieved by minimizing the divergence:

$$\mathcal{L}_\mathrm{reg} = \sum_{i=1}^{c} \hat{\boldsymbol{y}}_{\mathrm{I},i} \log \frac{\hat{\boldsymbol{y}}_{\mathrm{I},i}}{\boldsymbol{u}_i}, \tag{3}$$

where $\boldsymbol{u}_i = 1/c$ and $\hat{\boldsymbol{y}}_{\mathrm{I},i}$ corresponds the prediction results of $i$-th class.

By explicitly encouraging $\hat{\boldsymbol{y}}_\mathrm{I}$ toward a uniform distribution, the TIA space is discouraged from retaining informative features with respect to class labels. Synergistically, by combining the orthogonality loss $\mathcal{L}_\mathrm{ortho}$, the TRA subspace is motivated to concentrate on discriminative representations that are maximally aligned with the task objective. Consequently, the dual-stream representation achieves clearer task-oriented feature disentanglement and retains the task-relevant features in the TRA space.

### 3.5. Overall Objective

The full training objective integrates the task prediction loss with the two aforementioned components, which can be formulated as:

$$\mathcal{L} = \mathcal{L}_{\text{cls}} + \lambda_{\text{ortho}}\,\mathcal{L}_{\text{ortho}} + \lambda_{\text{reg}}\,\mathcal{L}_{\text{reg}}. \tag{4}$$

In the above equation, $\mathcal{L}_{\text{cls}}$ is the standard cross-entropy loss function for optimizing classification tasks. $\lambda_{\text{ortho}}$ and $\lambda_{\text{reg}}$ are hyperparameters that balance the contribution of the proposed orthogonality loss and distributional regularization loss. The overall training strategy is summarized in Algorithm 1.

---

**Algorithm 1:** Training and inference with DINOv3-FD

---
**Input:** Training set $\mathcal{D}_{\text{train}}$, test set $\mathcal{D}_{\text{test}}$
**Input:** Pretrained encoder $f_\theta$ with LoRA; adapters TRA and TIA; classification heads $g_\phi$ and $g_\psi$; dependency estimator $E_\omega$
**Input:** Hyperparameters $\lambda_{\text{ortho}}, \lambda_{\text{reg}}$

**Training phase:**
**for** *minibatch $(x, y) \subset \mathcal{D}_{train}$* **do**
    $\boldsymbol{f}_{\text{cls}} \leftarrow f_\theta(x)$;
    $\hat{\boldsymbol{y}}_{\text{T}} \leftarrow g_\phi(\text{TRA}(\boldsymbol{f}_{\text{cls}})), \quad \hat{\boldsymbol{y}}_{\text{I}} \leftarrow g_\psi(\text{TIA}(\boldsymbol{f}_{\text{cls}}))$;
    $\mathcal{L}_{\text{cls}} \leftarrow \text{CrossEntropy}(\hat{\boldsymbol{y}}_{\text{T}}, y)$
    $\mathcal{L}_{\text{reg}} \leftarrow \sum_{i=1}^{c} \hat{\boldsymbol{y}}_{\text{I},i} \log \frac{\hat{\boldsymbol{y}}_{\text{I},i}}{1/c}$;
    $(\boldsymbol{f}_{\text{T}}, \boldsymbol{f}_{\text{I}})_{\text{pair}} \leftarrow$ same-image tuples, $\quad (\boldsymbol{f}_{\text{T}}, \boldsymbol{f}_{\text{I}})_{\text{shuffle}} \leftarrow$ cross-image tuples;
    $\mathcal{L}_{\text{ortho}} \leftarrow \mathbb{E}_{(\boldsymbol{f}_{\text{T}}, \boldsymbol{f}_{\text{I}})_{\text{pair}} \sim \mathcal{D}_{T,I}}[E_\omega([\boldsymbol{f}_T, \boldsymbol{f}_I])] - \log \mathbb{E}_{(\boldsymbol{f}_{\text{T}}, \boldsymbol{f}_{\text{I}})_{\text{shuffle}} \sim \mathcal{D}_T \mathcal{D}_I}[\exp(E_\omega([\boldsymbol{f}_T, \boldsymbol{f}_I]))]$;
    $\mathcal{L} \leftarrow \mathcal{L}_{\text{cls}} + \lambda_{\text{ortho}}\,\mathcal{L}_{\text{ortho}} + \lambda_{\text{reg}}\,\mathcal{L}_{\text{reg}}$;
    Update TRA, TIA, $g_\phi$, $g_\psi$, $E_\omega$, and unfrozen PEFT parameters.;
**end**

**Inference phase:**
**for** *each image $x \in \mathcal{D}_{test}$* **do**
    $\boldsymbol{f}_{\text{cls}} \leftarrow f_\theta(x)$;
    $\hat{\boldsymbol{y}}_{\text{T}} \leftarrow g_\phi(\text{TRA}(\boldsymbol{f}_{\text{cls}}))$
    **return** $\hat{\boldsymbol{y}}_{\text{T}}$;
**end**

---

## 4. Experiments

### 4.1. Datasets and Preprocessing

We evaluate DINOv3-FD on three medical image classification tasks. The RSNA Pneumonia dataset(Anouk Stein et al., 2018) provides unique frontal-view chest radiographs labeled for pneumonia presence and follows a split of 21,346, 2,668, and 2,670 for training, validating, and testing. ISIC 2018(Codella et al., 2019; Tschandl et al., 2018) consists of dermoscopic images across seven diagnostic categories and adopts the official competition partition, with 10,015 cases for train, 193 cases for validation, and 1,512 cases for

test. ODIR-5K(ODIR) contains 5,000 color fundus photographs labeled for eight ocular conditions; the imagefolder format expands its multi-label structure into class-wise directories, resulting in 6,392 images due to multi-label duplication, as 5,110 are trained, 635 are validated and 647 are tested. Training images are augmented with scale-jittering and horizontal flipping, whereas validation and test images are resized to a fixed $256 \times 256$ and subsequently center-cropped to $224 \times 224$. All images are normalized using ImageNet statistics to ensure consistency with the original DINOv3 pretraining.

### 4.2. Implementation Details

**Training details**   All experiments are conducted on one NVIDIA RTX A6000 GPU. Models are trained using the ViT-L/16 DINOv3 encoder as the backbone, with its [CLS] representation serving as the global feature. Training is performed with AdamW optimizer using an initial learning rate of $5 \times 10^{-5}$, batch size of 16, and weight decay of $5 \times 10^{-2}$. The hyperparameter $\lambda_{\mathrm{ortho}}$ and $\lambda_{\mathrm{reg}}$ are set as 0.005 and 0.2.

**Metrics**   We report classification accuracy (ACC) and area under the ROC curve (AUC). For binary tasks such as RSNA, AUC is computed from the sigmoid probability of the positive class, and ACC uses a threshold of 0.5. For multi-class datasets (ISIC and ODIR), we employ a single softmax classifier and compute macro One-vs-Rest AUC along with top-1 ACC. We additionally report the macro $F_1$ score, which provides a balanced measure of per-class performance, especially for datasets with label imbalance.

## 5. Results

### 5.1. Comparisons with State-of-the-arts

In this section, we compare our DINOv3-FD with seven state-of-the-art (SOTA) PEFT approaches, including Linear Probe (Siméoni et al., 2025), Adapter-LN (Houlsby et al., 2019), LoRA (Hu et al., 2021), IA$^3$ (Liu et al., 2022), LyCORIS (Yeh et al., 2024), VeRA (Kopiczko et al., 2024), and PaCA (Woo et al., 2025). The quantitative results are presented in Table 1. As shown in the table, DINOv3-FD consistently achieves the best overall performance. For instance, it reaches 87.45%, 85.91%, and 73.11% accuracy on the three datasets, outperforming the second-best model by 0.56%, 3.11%, and 6.96%, respectively. Similar observations can be found in other evaluation metrics as well. Notably, our method exhibits the largest improvement on ODIR, which provides the smallest number of training samples, further demonstrating the effectiveness of our feature disentanglement mechanism in low-data regimes. Furthermore, while the compared PEFT methods exhibit fluctuating performance across datasets, DINOv3-FD remains consistently superior, demonstrating its robustness under diverse medical tasks. Meanwhile, we include non-GVFM baselines trained from scratch with standard ResNet50 and ViT-B architectures. The inferior performance in Table 1 indicates that heavy vision models cannot be effectively trained from scratch under a small-scale medical dataset.

Table 1: Quantitative performance in percentage (%) across RSNA, ISIC, and ODIR datasets. The best and second-best performances are marked in **bold** and underline.

| Method | RSNA (%) | | | ISIC (%) | | | ODIR (%) | | |
|---|---|---|---|---|---|---|---|---|---|
| | ACC | AUC | $F_1$ | ACC | AUC | $F_1$ | ACC | AUC | $F_1$ |
| ResNet50 | 84.61 | 87.85 | 63.72 | 64.62 | 85.33 | 32.73 | 48.22 | 72.84 | 24.55 |
| ViT-B | 83.33 | 83.66 | 53.60 | 63.76 | 81.96 | 23.39 | 44.51 | 52.50 | 7.71 |
| Linear Probe | 80.97 | 81.27 | 36.34 | 57.41 | 87.04 | 40.65 | 31.38 | 72.82 | 35.27 |
| Adapter-LN | 86.89 | 90.66 | 67.29 | 78.57 | 96.45 | 70.10 | 56.57 | 88.84 | 55.75 |
| LoRA | 86.33 | 91.08 | 66.91 | 82.28 | 96.70 | 72.43 | 65.84 | 91.40 | 65.45 |
| IA$^3$ | 86.63 | 90.22 | 67.04 | 82.80 | 96.38 | 70.19 | 66.15 | 88.80 | 56.03 |
| LyCORIS | 84.12 | 87.18 | 60.00 | 79.70 | 94.17 | 65.29 | 62.60 | 87.15 | 50.82 |
| VeRA | 85.09 | 88.16 | 62.02 | 79.03 | 94.68 | 63.37 | 60.90 | 86.58 | 49.87 |
| PaCA | 86.70 | 91.09 | 68.83 | 81.61 | 96.48 | 72.47 | 58.57 | 90.23 | 58.33 |
| **Ours** | **87.45** | **91.39** | **70.06** | **85.91** | **96.99** | **75.17** | **73.11** | **92.00** | **65.59** |

## 5.2. Ablation Studies

**Effect of key components.** We evaluate the contribution of key components in our DINOv3-FD through a set of ablation studies, where each component is progressively introduced to assess its impact on the overall performance. The setting configurations include: (1) Removing $\mathcal{L}_{\text{ortho}}$ and $\mathcal{L}_{\text{reg}}$, which degenerates to the Lora-based fine-tuning. (2) With $\mathcal{L}_{\text{ortho}}$. (3) With $\mathcal{L}_{\text{reg}}$, which equals to our DINOv3-FD with all components. The results are presented in Table 2. As shown, incorporating the orthogonality loss consistently improves performance across all evaluated tasks. Moreover, further integrating the distributional regularization loss yields additional gains and leads to stronger overall results.

Table 2: Ablation study with orthogonality and distributional regularization loss.

| $\mathcal{L}_{\text{ortho}}$ | $\mathcal{L}_{\text{reg}}$ | RSNA (%) | | | ISIC (%) | | | ODIR (%) | | |
|---|---|---|---|---|---|---|---|---|---|---|
| | | ACC | AUC | $F_1$ | ACC | AUC | $F_1$ | ACC | AUC | $F_1$ |
| ✗ | ✗ | 86.33 | 91.08 | 66.91 | 82.28 | 96.70 | 72.43 | 65.84 | 91.40 | 65.45 |
| ✓ | ✗ | 86.44 | 91.34 | 65.78 | 83.53 | **97.07** | **76.02** | 66.31 | 90.92 | 65.45 |
| ✓ | ✓ | **87.45** | **91.39** | **70.06** | **85.91** | 96.99 | 75.17 | **73.11** | **92.00** | **65.59** |

**Comparison with alternative regularization objectives.** In this section, we compare our method with alternative modules in terms of the distributional regularization loss. Our method adopts a Kullback–Leibler divergence (KLD) for promoting the output of TIA toward a uniform distribution. We compare KLD with two other alternative modules, GRL and ER. Specifically, Gradient reversal (GRL) (Ganin et al., 2016) suppresses class-dependent signals through an adversarial objective and performs well in settings where

the feature structure aligns with such inverted gradients. Entropy-based regularization (ER) (Grandvalet and Bengio, 2004) provides a softer constraint by encouraging high-entropy predictions. The quantitative results are shown in Table 3. Across all datasets, our method achieves the best and stable performance. Enforcing a uniform predictive distribution within TIA, it provides more explicit targets over the irrelevant pathway and complements the orthogonality loss in constructing a disentangled representation.

Table 3: Ablation study with alternative distributional regularization objectives

| Methods | RSNA (%) | | | ISIC (%) | | | ODIR (%) | | |
|---|---|---|---|---|---|---|---|---|---|
| | ACC | AUC | $F_1$ | ACC | AUC | $F_1$ | ACC | AUC | $F_1$ |
| GRL | 87.37 | **91.68** | 69.50 | 78.04 | 95.81 | 72.66 | 55.49 | 87.37 | 51.83 |
| ER | 87.42 | 91.33 | 68.72 | 83.86 | **97.06** | 68.44 | 67.85 | 87.73 | 53.25 |
| KLD (Ours) | **87.45** | 91.39 | **70.06** | **85.91** | 96.99 | **75.17** | **73.11** | **92.00** | **65.59** |

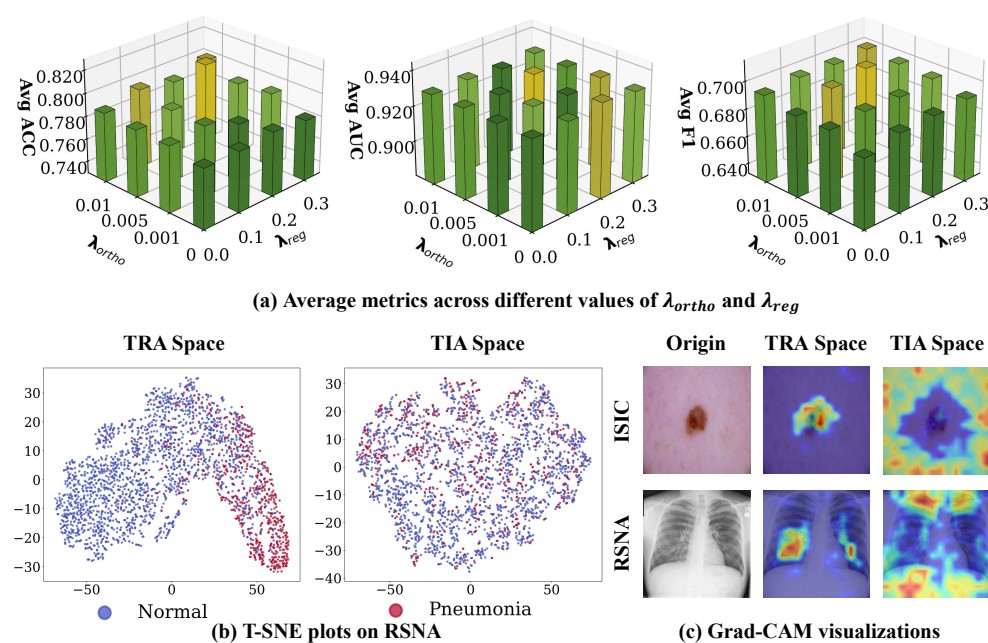

(a) Average metrics across different values of $\lambda_{ortho}$ and $\lambda_{reg}$

(b) T-SNE plots on RSNA

(c) Grad-CAM visualizations

Figure 2: (a) 3D bar plot of average ACC, AUC and $F_1$ scores across three datasets under different values of $\lambda_{\mathrm{ortho}}$ and $\lambda_{\mathrm{reg}}$. Warmer (more yellow) colors indicate higher performance. (b) T-SNE visualizations of the feature embeddings from task-relevant and task-irrelevant adapters over RSNA. (c) Grad-CAM visualization maps of task-relevant and task-irrelevant pathways of two randomly selected samples on ISIC and RSNA datasets.

**Effect of loss weights.** To evaluate the sensitivity of our framework to the hyperparameters $\lambda_{\mathrm{ortho}}$ and $\lambda_{\mathrm{reg}}$, we assess the average performance across three datasets under

different configurations. As illustrated in Figure 2-(a), compared with fully removing the feature disentanglement mechanism ($\lambda_{\mathrm{ortho}} = 0$, $\lambda_{\mathrm{reg}} = 0$), the performance consistently improves across various values of $\lambda_{\mathrm{ortho}}$ and $\lambda_{\mathrm{reg}}$ in terms of three metrics averaged on three datasets. The best trade-off is achieved at $\lambda_{\mathrm{ortho}} = 0.005$ and $\lambda_{\mathrm{reg}} = 0.2$. We adopt this configuration as the default setting for our method.

### 5.3. Interpretable Visualization

**Feature embedding visualization.** We utilize the t-SNE (t-distributed Stochastic Neighbor Embedding) method to visualize the feature embeddings from the two branches of our DINOv3-FD. The result over RSNA dataset is depicted in Figure 2-(b). As shown, the left and right regions correspond to the TRA and TIA feature spaces, respectively, with different colors denoting different disease categories. In the TRA space, normal and pneumonia samples form two clearly separated clusters, indicating strong task relevance. In contrast, the TIA space exhibits a highly random distribution with no visible class separation. These visualization results demonstrate the effectiveness of our feature disentanglement mechanism and validate that TRA captures task-relevant features. More visualization results can be found in the Appendix.

**Feature localization visualization.** To further locate the regions of interest of the two branches during classification, we use Grad-CAM (Selvaraju et al., 2017) to generate class-specific activation maps for visualization. Figure 2-(c) illustrates the activation patterns of two randomly selected samples from RSNA and ISIC. As shown, the TRA branch focuses on clinically meaningful regions that support the decision-making process. For example, TRA highlights lesion areas in dermatology images and lung regions in chest X-ray images. In contrast, the TIA branch predominantly attends to regions with limited diagnostic relevance, including background or nonspecific structures. These qualitative observations further validate the complementary nature of the two branches and demonstrate the success of our feature disentanglement strategy. More visualization results can be found in the Appendix.

## 6. Conclusion

In this work, we proposed DINOv3-FD, a feature disentanglement framework that enables parameter-efficient adaptation of DINOv3 for medical vision tasks. By leveraging a dual adapter framework, we disentangle the task-relevant and task-irrelevant representations into two subspaces. Furthermore, the proposed orthogonality and distributional regularization objectives further promote the disentanglement procedure in a task-oriented manner. Extensive experiments across multiple medical datasets demonstrate that DINOv3-FD outperforms other existing PEFT approaches. In the future, we plan to explore this framework in three directions. First, we will incorporate textual information such as radiology reports and clinical metadata to enable text-guided adaptation, allowing the model to exploit cross-modal interaction during adaptation. Second, we will explore how to adapt DINOv3, which is trained on 2D images, to 3D medical imaging modalities, like CT and MRI volumes. Third, we will further evaluate the framework on a wider range of vision tasks, such as segmentation and lesion localization.

## Acknowledgments

This work was supported by Tier 1 grant, NUS, Singapore (24-1250-P0001) and Ministry of Education Tier 2 grant, Singapore (T2EP20224-0028).

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

# Appendix A. Additional Embedding and Localization Visualizations

We further explored DINOv3's powerful representation skills across all three datasets.

## A.1. T-SNE Analyses on ISIC and ODIR

To further examine the representational behavior of DINOv3-FD beyond the RSNA dataset, we provide additional t-SNE embeddings on ISIC and ODIR. As illustrated in Figure 3-(a), the task-relevant adapter (TRA) retains well-structured and semantically aligned clusters across both dermatology (ISIC) and ophthalmology (ODIR) domains. Disease categories with distinct visual signatures, such as basal cell carcinoma and glaucoma, form clearly separated regions, highlighting the adapter's ability to consolidate diagnostic cues.

In contrast, the task-irrelevant adapter (TIA) produces highly intermixed distributions with no meaningful class separation, reinforcing its role as a repository for non-discriminative or nuisance factors. These complementary behaviors are consistent with our design goal and match the observations reported in the main text.

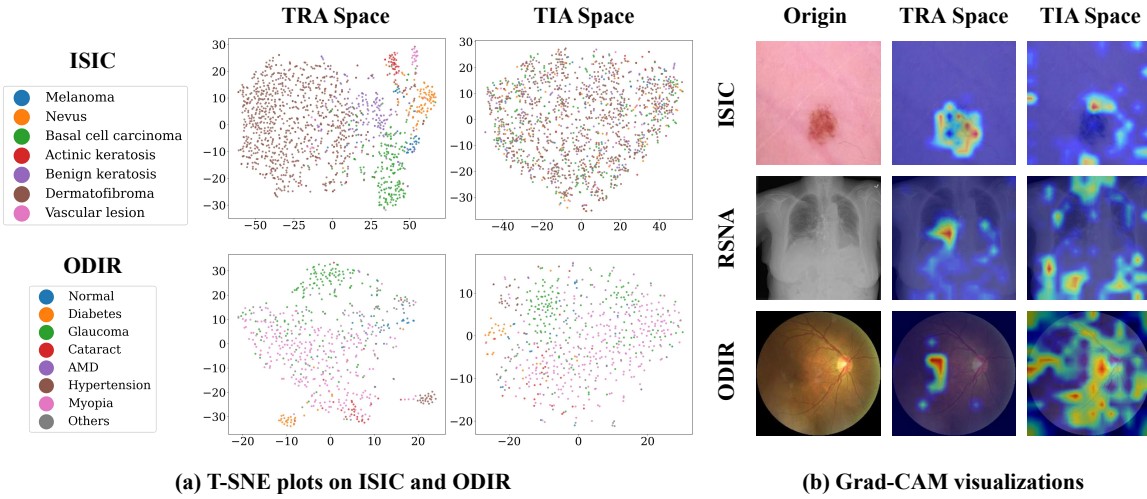

(a) T-SNE plots on ISIC and ODIR            (b) Grad-CAM visualizations

Figure 3: **(a) T-SNE plots on ISIC and ODIR.** The TRA space (left of each pair) forms compact and semantically aligned clusters, whereas the TIA space (right) exhibits diffuse and unstructured patterns. Colors correspond to clinical categories as defined in the datasets. **(b) Grad-CAM visualizations on ISIC, RSNA, and ODIR.** The TRA branch (middle) consistently concentrates on disease-relevant regions, while the TIA branch (right) focuses on non-diagnostic structures. These results validate the complementary representational roles introduced by our disentanglement framework.

## A.2. Grad-CAM Localization Across Three Datasets

We additionally report Grad-CAM visualizations over ISIC, RSNA, and ODIR to further probe the spatial focus of the two adapters. As shown in Figure 3-(b), the TRA pathway consistently highlights clinically informative regions, including dermoscopic lesions, pulmonary opacities, and disease-related fundus structures. The TIA pathway, by contrast, attends to diffuse or anatomically non-salient areas, often favoring background textures or peripheral

Table 4: Stability analysis across different random seeds (0, 1, 42). We report mean $\pm$ standard deviation for all metrics.

| Method | Dataset | ACC | AUC | F1 |
|--------|---------|-----|-----|-----|
| LoRA | RSNA | 86.00±0.5 | 90.67±0.6 | 66.57±0.9 |
| | ISIC | 80.34±1.9 | 96.68±0.3 | 73.33±0.8 |
| | ODIR | 65.43±0.9 | 90.84±0.5 | 64.59±0.8 |
| Ours | RSNA | 86.27±1.0 | 90.72±0.7 | 66.87±2.8 |
| | ISIC | 84.41±1.5 | 96.90±0.3 | 73.35±2.3 |
| | ODIR | 72.33±0.8 | 91.57±0.7 | 65.71±2.2 |

structures. This complementary spatial behavior reflects the intended disentanglement between task-discriminative and task-agnostic information, offering an interpretable view of how DINOv3-FD separates diagnostic cues from incidental image content.

## Appendix B. Stability Analysis

### B.1. Stability Across Different Seeds and Batch Sizes

We further analyze training stability by repeating experiments with different random seeds. As summarized in Table 4, DINOv3-FD exhibits low variance across runs, indicating that the proposed dependence-minimization objective can be optimized reliably. We also compare against a representative baseline, LoRA, observing consistently better performance.

Besides multiple seed runs, we also conduct experiments with the batch size of 16, 32, and 64. As Table 5 shows, although there is slight turbulence with changes in batch size, we notice that the batch size of 16 achieves the best average performance across 3 datasets. Notably, all these 3 options outperform other PEFT baselines, demonstrating the effectiveness of feature disentanglement.

Table 5: Effect of batch size on performance across RSNA, ISIC, and ODIR datasets. The best and second-best performances are marked in **bold** and underline.

| Batch Size | RSNA (%) | | | ISIC (%) | | | ODIR (%) | | |
|------------|----------|-----|-----|----------|-----|-----|----------|-----|-----|
| | ACC | AUC | F1 | ACC | AUC | F1 | ACC | AUC | F1 |
| 64 | 86.67 | 91.34 | 66.85 | 83.00 | 96.52 | 70.78 | 71.25 | 91.56 | 63.87 |
| 32 | 86.40 | 91.24 | **70.27** | 84.39 | **97.19** | **75.24** | 72.33 | 91.86 | **66.26** |
| 16 (Ours) | **87.45** | **91.39** | 70.06 | **85.91** | 96.99 | 75.17 | **73.11** | **92.00** | 65.59 |

### B.2. Comparison with alternative orthogonal objectives

We compare the proposed dependence-minimization objective with simpler decorrelation losses like cross-covariance (Bardes et al., 2021), cosine decorrelation (Zbontar et al., 2021),

and Gram (Cogswell et al., 2015), and even more complex losses like HSIC (Ma et al., 2020). The results are shown in Table 6. While these alternatives show distinct performance across different datasets, our choice has the highest average outcome among them.

Table 6: Comparison of different decorrelation objectives across RSNA, ISIC, and ODIR datasets. The best and second-best performances are marked in **bold** and underline.

| Method | RSNA (%) | | | ISIC (%) | | | ODIR (%) | | |
|---|---|---|---|---|---|---|---|---|---|
| | ACC | AUC | F1 | ACC | AUC | F1 | ACC | AUC | F1 |
| HSIC | 86.97 | 91.34 | 70.00 | 85.58 | 96.89 | 76.42 | 71.25 | 91.70 | 66.05 |
| Gram | 86.63 | 91.33 | 68.98 | 85.65 | 96.91 | 75.98 | 72.33 | **92.42** | **67.22** |
| Cross Covariance | 86.74 | 91.08 | 66.60 | **88.60** | **98.89** | **77.38** | 72.64 | 91.51 | 65.52 |
| Cosine | 87.23 | **91.45** | 68.22 | 85.58 | 96.79 | 74.95 | 71.87 | 91.51 | 65.17 |
| **MINE (Ours)** | **87.45** | 91.39 | **70.06** | 85.91 | 96.99 | 75.17 | **73.11** | 92.00 | 65.59 |

## Appendix C. Strategies to Ease Class Imbalance Settings

### C.1. Effect of Imbalance-handling Strategies

Given the fact that RSNA, ISIC, and ODIR have an imbalanced distribution, we conduct evaluation experiments with two approaches: one is using a balanced data sampler, and the other is using reweighting, to investigate their effectiveness. As Table 7 shows, our original setup shows a better average performance, demonstrating the effectiveness of our method.

Table 7: Effect of imbalance-handling strategies on RSNA, ISIC, and ODIR datasets. The best and second-best performances are marked in **bold** and underline.

| Method | RSNA (%) | | | ISIC (%) | | | ODIR (%) | | |
|---|---|---|---|---|---|---|---|---|---|
| | ACC | AUC | F1 | ACC | AUC | F1 | ACC | AUC | F1 |
| Balanced Sampling | 79.63 | **91.49** | 66.13 | 84.72 | **97.35** | **77.27** | 59.66 | 90.47 | 58.33 |
| Reweighting | 86.97 | 91.42 | 68.01 | 85.38 | 96.87 | 75.43 | 71.87 | 90.82 | **66.70** |
| **Ours** | **87.45** | 91.39 | **70.06** | **85.91** | 96.99 | 75.17 | **73.11** | **92.00** | 65.59 |

### C.2. Comparison with Prior Matching.

While we regularize the task-irrelevant adapter toward a uniform distribution by default, we also evaluate a prior-matching variant (Bhat et al., 2025) that aligns predictions with the empirical class distribution to check if it is better for an imbalance situation. The results shown in Table 8 indicate that our method mostly performs better than the prior-matching approach. This implies that enforcing a uniform target can effectively remove task-irrelevant information from the task-relevant features.

Table 8: Comparison between prior-matching regularization and uniform regularization. The better performances are marked in **bold**.

| Method | RSNA (%) | | | ISIC (%) | | | ODIR (%) | | |
|---|---|---|---|---|---|---|---|---|---|
| | ACC | AUC | F1 | ACC | AUC | F1 | ACC | AUC | F1 |
| Prior Matching | 86.67 | 91.21 | 69.26 | 85.58 | 96.88 | **75.92** | 71.72 | 90.80 | 64.96 |
| **Uniform (Ours)** | **87.45** | **91.39** | **70.06** | **85.91** | **96.99** | 75.17 | **73.11** | **92.00** | **65.59** |

## Appendix D.  Transferability to Segmentation Tasks

To evaluate whether the task-relevant adapter (TRA) captures clinically meaningful and transferable representations beyond classification, we further assess DINOv3-FD on a dense prediction task. Specifically, we transfer the learned TRA features to the ISIC 2018 Lesion Boundary Segmentation task.

As shown in Table 9, DINOv3-FD consistently outperforms representative PEFT baselines. In particular, our method achieves a Dice score of **91.26%** and an IoU of **83.93%**, surpassing all competing methods. These results indicate that the proposed method can generalize to dense visual prediction tasks.

Table 9: Quantitative performance in percentage (%) on the ISIC 2018 Lesion Boundary Segmentation task. The best and second-best performances are marked in **bold** and underline.

| Method | ACC (%) | Dice (%) | IoU (%) | mAP (%) |
|---|---|---|---|---|
| Linear Probe | 38.57 | 46.53 | 30.32 | 35.86 |
| Adapter-LN | 91.65 | 90.67 | 82.94 | 98.71 |
| LoRA | 92.12 | 91.25 | 83.91 | 98.91 |
| IA$^3$ | 91.71 | 90.64 | 82.88 | 98.66 |
| LyCORIS | 91.27 | 90.07 | 81.94 | 98.25 |
| VeRA | 91.40 | 90.22 | 82.19 | 98.32 |
| PaCA | 91.93 | 90.85 | 83.23 | 98.72 |
| **Ours** | **92.14** | **91.26** | **83.93** | **98.91** |

## Appendix E.  Efficiency Analysis

We also calculate the computational overhead as shown in Table 10. Our method accounts for around 4.7M trainable parameters, which is comparable to LoRA. The GFLOPs is 123.11, which is comparable to other PEFT baselines.

Table 10: Performance of efficiency metrics on all baselines and our method. The best and second-best performances are marked in **bold** and underline. Param. stands for learnable parameters.

| Method | Efficiency | |
|---|---|---|
| | Param. | GFLOPs |
| ResNet50 | 26.4M | 8.18 |
| ViT-B | 30.3M | 33.70 |
| Linear Probe | 0.3M | 121.76 |
| Adapter-LN | 1.7M | 121.76 |
| LoRA | 4.7M | 123.11 |
| IA$^3$ | 1.8M | 121.80 |
| LyCORIS | 1.6M | 121.76 |
| VeRA | 1.8M | 123.07 |
| PaCA | 3.3M | 121.76 |
| **Ours** | 4.7M | 123.11 |

