# OpenReview forum: "Incentivizing DINOv3 Adaptation for Medical Vision Tasks via Feature Disentanglement"
_MIDL.io/2026/Conference — MIDL 2026 Poster_

### Official Review · Reviewer_pzKj · 2026-01-06

**Confidence:** 4
**Preliminary Rating:** 4

**Summary:**

This paper proposes DINOv3-FD, a feature disentanglement framework for parameter-efficient fine-tuning (PEFT) of the DINOv3 vision foundation model on medical imaging tasks. Evaluated on three medical classification datasets, DINOv3-FD outperforms existing PEFT methods, demonstrating the effectiveness of its task-oriented disentanglement approach for robust adaptation in medical domains.

**Strengths:**

+ Medical image classification aligns well with the theme of MIDL.
+ The application of DINOv3 effectively reflect the latest advancements in the field.
+ The paper is generally well-written and easy to follow.
+ Extensive experiments on multiple datasets demonstrate the superiority of the proposed DINOv3-FD.
+ The codes are open available.

**Weaknesses:**

- It is recommended to include the number of learnable parameters for different methods in Table 1.
- More feature disentanglement-based medical image analysis methods, including [*1,*2], should be added in Section 2.2.

[*1] Disentangling disease-related representation from obscure for disease prediction
[*2] Semi-and Weakly-Supervised Learning for Mammogram Mass Segmentation with Limited Annotations

**Detailed Comments:**

See weaknesses.

**Justification Of The Preliminary Rating:**

Good novelty and thorough experiments. See strengths:
+ Medical image classification aligns well with the theme of MIDL.
+ The application of DINOv3 effectively reflect the latest advancements in the field.
+ The paper is generally well-written and easy to follow.
+ Extensive experiments on multiple datasets demonstrate the superiority of the proposed DINOv3-FD.
+ The codes are open available.

**Questions To Address In The Rebuttal:**

See weaknesses.

---

> ### Author Response · Authors · 2026-01-25
> **Response to Reviewer pzKj**
>
> (i) Learnable parameters
>
> Thanks for your valuable suggestion. In order to adhere to the official MIDL LaTeX template, we have added the number of learnable parameters for all methods in the appendix to avoid altering font size within tables. As shown in Table 1, our method introduces 4.7M trainable parameters, which is comparable to LoRA and only accounts for approximately 1.5% of the backbone parameters.
> |Method|Trainable Parameters|
> |-|-|
> |Linear Probe|0.3M|
> |Adapter-LN|1.7M|
> |LoRA|4.7M|
> |IA³|1.8M|
> |LyCORIS|1.6M|
> |VeRA|1.8M|
> |PaCA|3.3M|
> |Ours|4.7M|
>
> (ii) Related work on disentanglement-based medical imaging
>
> We thank the reviewer for the suggestions. We will include the recommended works and expand Section 2.2 to provide a more comprehensive comparison with prior medical feature disentanglement literature. We also refined the reference list to remove errors.

---

### Official Review · Reviewer_Vg8X · 2026-01-07

**Confidence:** 3
**Preliminary Rating:** 4

**Summary:**

This paper propose DINOv3-FD, a task-oriented feature disentanglement framework that adapts DINOv3 to medical vision tasks. DINOv3-FD introduces a dual-stream adapter that separates features into task-relevant and task-irrelevant subspaces, reinforced by an orthogonality loss to encourage their mutual independence. Experimental results show that DINOv3-FD outperforms other PEFT strategies over three medical classification tasks, demonstrating the effectiveness of feature disentanglement.

**Strengths:**

The concept is novel and well-motivated. The central idea of explicitly disentangling task-relevant from task-irrelevant features during adaptation is highly relevant and under-explored in the context of adapting general vision foundation models (GVFMs) to medical imaging. The motivation is clear: pretrained features from natural images likely contain a mixture of signals, and selectively enhancing the relevant ones is a promising direction for improving performance and robustness. The paper is well-written. The codes are available.

**Weaknesses:**

While the paper emphasizes parameter efficiency, it would be beneficial to include a brief comparison of the actual computational cost (e.g., training time, inference time, or FLOPs) of DINOv3-FD against the baselines. Since DINOv3-FD introduces two adapters and a dependency estimator, a discussion on the trade-off between added complexity and performance gain would be valuable for practitioners.

**Detailed Comments:**

n/a

**Justification Of The Preliminary Rating:**

Overall, the method is evaluated on three medical datasets and demonstrates superior performance compared to several state-of-the-art PEFT baselines. The paper is well-written, the methodology is sound and motivated by a clear problem statement, and the experimental section is comprehensive.

**Questions To Address In The Rebuttal:**

While the paper emphasizes parameter efficiency, it would be beneficial to include a brief comparison of the actual computational cost (e.g., training time, inference time, or FLOPs) of DINOv3-FD against the baselines. Since DINOv3-FD introduces two adapters and a dependency estimator, a discussion on the trade-off between added complexity and performance gain would be valuable for practitioners.

---

> ### Author Response · Authors · 2026-01-25
> **Response to Reviewer Vg8X: Efficiency and computational cost**
>
> We thank the reviewer for highlighting this practical consideration. Our method introduces two lightweight adapters and a small dependence estimator, which does not introduce significant computational overhead. Specifically, as shown in Table 1, the total number of trainable parameters is 4.7M, which is comparable with LoRA. The inference cost is 25.32 ms per image (batch size = 1), corresponding to 123.11 GFLOPs, which is comparable to other PEFT baselines.
> |Method|Trainable Parameters|Inference Time|GFLOPs|
> |-|-|-|-|
> |Linear Probe|0.3M|25.06ms|121.76|
> |Adapter-LN|1.7M|25.08ms|121.76|
> |LoRA|4.7M|30.65ms|123.11|
> |IA³|1.8M|26.08ms|121.80|
> |LyCORIS|1.6M|25.07ms|121.76|
> |VeRA|1.8M|31.33ms|123.07|
> |PaCA|3.3M|38.56ms|121.76|
> |Ours|4.7M|25.32ms|123.11|

---

### Official Review · Reviewer_iVWF · 2026-01-10

**Confidence:** 4
**Preliminary Rating:** 4
**Final Rating:** 5

**Summary:**

This paper proposes a representation-disentanglement framework for adapting general vision foundation models to medical imaging. The key idea is to explicitly separate task-relevant and task-irrelevant factors through a dual-stream design, and to enforce independence between the two streams using an “orthogonality” objective based on contrasting the joint distribution with a shuffled product-of-marginals. The approach is complemented by a regularization that encourages the irrelevant branch to be non-informative. The method is evaluated on multiple medical classification benchmarks with comparisons against several PEFT baselines and with ablation studies.

**Strengths:**

* Clear and motivated idea. The disentanglement perspective is intuitive and well aligned with the domain-gap problem in medical imaging. The formulation of the dependence-minimization objective is interesting and goes beyond standard feature decorrelation heuristics.

* Empirical evidence is solid for the chosen setting. The comparisons to competing methods and the ablation studies are well organized and collectively support the effectiveness of the proposed components.

* Reproducibility. The authors provide code to reproduce the results, which is valuable for the community.

**Weaknesses:**

* Limited task coverage (classification only). The evaluation focuses exclusively on classification tasks. Extending experiments to segmentation or detection would strengthen the manuscript because these tasks are often more interpretable and can better demonstrate whether the “relevant” branch captures clinically meaningful structures rather than dataset-specific shortcuts.

* Missing “from-scratch / non-GVFM” downstream SOTA baselines. To convincingly demonstrate the value of transferring knowledge from large-scale natural images, it would be helpful to report strong task-specific baselines trained without GVFM initialization (or at least well-established SOTA reported in the medical imaging literature) alongside the fine-tuned GVFM results. This would contextualize how much improvement is attributable to the foundation model pretraining versus the proposed adaptation strategy.

* Stability of the dependence-minimization (“orthogonality”) loss. Although described as an orthogonality constraint, the proposed loss is implemented via a mutual-dependence minimization objective using a Donsker–Varadhan style estimator that contrasts samples from the joint distribution and a shuffled product-of-marginals. Such MI-style estimators can be sensitive to batch size, the capacity of the estimator network, and optimization dynamics, and the paper does not currently report variance across random seeds or other stability diagnostics.

* Interaction with severe class imbalance / rare diseases is unclear. Medical datasets frequently exhibit long-tailed label distributions where rare classes are extremely sparse. Under this condition, the shuffled negative sampling used to approximate product-of-marginals may become noisy (especially when a minibatch contains very few rare-class examples), potentially yielding high-variance gradients biased toward majority-class statistics. In addition, the regularization that pushes the irrelevant head toward a uniform predictive distribution may introduce conflicting gradients under heavy imbalance (since a uniform target can be far from the empirical class prior), and could disproportionately harm minority-class separability. The manuscript would be stronger with explicit analysis of this regime.

**Detailed Comments:**

**Beyond classification:** Can the authors evaluate the method on at least one segmentation or detection benchmark to demonstrate that the relevant branch learns clinically meaningful structure and that the approach generalizes across task types?

**Contextual baselines:** Please include (or cite and clearly report) strong downstream SOTA baselines trained without GVFM pretraining, so the added value of transferring from natural-image foundation models is transparent.

**Stability analysis:** How stable is the dependence-minimization estimator across random seeds and different batch sizes? Reporting mean ± std over multiple seeds would improve confidence. Have the authors compared against simpler independence penalties (e.g., cross-covariance / cosine decorrelation) as alternative objectives?

**Imbalance / long-tail evaluation:** Do the authors use class-balanced sampling or loss reweighting? Please report per-class or imbalance-sensitive metrics (macro-F1, macro-AUC, PR-AUC) and consider an ablation under controlled long-tail splits. It would also be interesting to test whether replacing “uniform” regularization with prior-matching (matching the empirical class prior) changes minority-class performance.

**Justification Of Final Rating:**

The authors have provided sufficient evidence in their extended experiments to show the effectiveness of the proposed method and fully addressed my concerns. I think the current manuscript is solid enough for more insight discussion in the conference.

**Justification Of The Preliminary Rating:**

Overall, the paper presents an appealing and well-supported approach to representation disentanglement for medical adaptation, with strong empirical results and good reproducibility. Addressing the above concerns—especially broader task evaluation and a clearer analysis of stability and long-tailed label distributions—would significantly strengthen the manuscript and make the conclusions more convincing for real-world medical deployments.

**Questions To Address In The Rebuttal:**

Please refer to the Weaknesses and concerns and Questions for the authors / suggestions sections above. In particular, the rebuttal should directly address: (i) the limitation of classification-only evaluation and whether evidence on segmentation/detection can be provided; (ii) the inclusion of strong non-GVFM downstream baselines to contextualize gains; (iii) the training stability and sensitivity of the dependence-minimization (“orthogonality”) estimator (e.g., across seeds and batch sizes); and (iv) the method’s behavior under severe class imbalance / long-tailed label distributions, including appropriate per-class or imbalance-sensitive metrics.

---

> ### Author Response · Authors · 2026-01-25
> **Response to Reviewer iVWF: Due to space limitation, we mainly show best performance on RSNA here, all experiment results will be included in the revised manuscript.**
>
> (i) Beyond classification: segmentation/detection
>
> We thank the reviewer for this constructive feedback. We agree that extending the evaluation beyond classification is important. Following the reviewer’s suggestions, we evaluate our method on the ISIC segmentation benchmark. The quantitative results are shown in Table 1. The results show that our method consistently improves Dice and IoU over other compared methods, indicating that our method can generalize to dense visual prediction tasks. We will include all results in the appendix.
> |Method|ACC (%)|Dice (%)|IoU (%)|mAP (%)|
> |-|-|-|-|-|
> |LoRA|92.12|91.25|83.91|98.91|
> |Ours|**92.14**| **91.26**|**83.93**|**98.91**|
>
> (ii) Non-GVFM / from-scratch baselines
>
> We thank the reviewer for this valuable suggestion. To investigate the gains from GVFM adaptation, we include non-GVFM baselines trained from scratch using standard ResNet50 and ViT-B architectures on the same datasets. As Table 2 shows, models trained from scratch achieve lower performance compared to GVFM-based adaptation, confirming that GVFM provides a strong foundation, especially for macro AUC and F1, and our strategy yields additional gains. Notably, when training ViT-L from scratch, we observe inferior performance, which indicates that heavier vision models cannot be effectively trained from scratch under a small-scale medical dataset. The whole comparison will be included in the paper.
> |Method|ACC (%)|AUC (%)|F1 (%)|
> |-|-|-|-|
> |ResNet50|84.61|87.85|63.72|
> |Ours|**87.45**|**91.39**|**70.06**|
>
> (iii) Stability of the dependence-minimization loss
>
> We follow the reviewer’s suggestion and conduct experiments as follows. All results will be shown in the appendix.
>
> Firstly, to assess stability, we repeat experiments under multiple seeds (0, 1, 42) and report the mean value and standard deviation of all metrics across datasets. Table 3 shows low variance across seeds, indicating stable optimization behavior. We also compare against a representative baseline, LoRA,  observing consistently better performance.
> |Method|ACC (%)|AUC (%)|F1 (%)|
> |-|-|-|-|
> |LoRA|86.00$\pm$0.5|90.67$\pm$0.6|66.57$\pm$0.9|
> |Ours|86.27$\pm$1.0|90.72$\pm$0.7|66.87$\pm$2.8|
>
> Secondly, we conduct experiments with the batch size of 16, 32, and 64. As Table 4 shows, although there is slight turbulence with changes in batch size, we notice that the batch size of 16 achieves the best average performance across 3 datasets. Notably, all these 3 options outperform other PEFT baselines, demonstrating the effectiveness of feature disentanglement.
> |Batch Size|ACC (%)|AUC (%)|F1 (%)|
> |-|-|-|-|
> |32|86.40|91.24|**70.27**|
> |16 (Ours)|**87.45**|**91.39**|70.06|
>
> Finally, we compare the proposed dependence-minimization objective with simpler decorrelation losses like cross-covariance, cosine decorrelation, and Gram, and even more complex losses like HSIC. Part of the results are shown in Table 5. While these alternatives show distinct performance across different datasets, our choice has the highest average outcome among them. In the future, we will explore a dynamic feature disentanglement strategy that adaptively tailors disentangling objectives on different datasets.
> |Method|ACC (%)|AUC (%)|F1 (%)|
> |-|-|-|-|
> |Cosine|87.23|**91.45**|68.22|
> |MINE (Ours)|**87.45**|91.39|**70.06**|
>
> (iv) Class imbalance / long-tailed labels
>
> Thanks for your valuable feedback. Actually, we already report macro-F1 and macro-AUC for multi-category datasets (ISIC, ORID), which are less sensitive to class frequency and better reflect minority-class behavior. For RSNA, as it only has two categories, we report standard binary F1 and AUC. Regarding the severity of class imbalance, we do notice that these 3 datasets have an imbalanced distribution. For example, two dominant classes in ODIR cover around 70%. Following the valuable suggestion, we conduct experiments with two approaches, i.e., using a balanced data sampler and using reweighting, to investigate their effectiveness. As Table 6 shows, our original setup shows an average better performance, demonstrating the effectiveness of our method. Moreover, considering our main scope is effective GVFM adaptation, all the methods in these papers undergo the same data splitting for fair comparison without handling imbalance. In the future, we will explore more effective adaptation ways for imbalanced data.
>
> Finally, we empirically evaluate the suggested prior-matching variant by replacing the uniform regularization with a KL objective that matches the empirical class prior. The results shown in Table 6 indicate that our method mostly performs better than the prior-matching approach. This implies that enforcing a uniform target can effectively remove task-irrelevant information from the task-relevant features. Other results will be included in the appendix.
> |Method|ACC (%)|AUC (%)|F1 (%)|
> |-|-|-|-|
> |Sampling|79.63|**91.49**|66.13|
> |Reweighting|86.97|91.42|68.01|
> |Prior|86.67|91.21|69.26|
> |Ours|**87.45**|91.39|**70.06**|

---

> > ### Comment · Reviewer_iVWF · 2026-01-30
> >
> > Thank you for the response. The supplimental results look promissing to me.

---

> > > ### Author Response · Authors · 2026-01-31
> > >
> > > We sincerely thank the reviewer for the valuable suggestions, which have greatly helped us improve the quality and clarity of our manuscript.

---

### Author Rebuttal · Authors · 2026-01-25

**Rebuttal:**

We thank the reviewers for their careful reading and constructive feedback. Below, we address the concerns point by point. All changes and clarifications will be incorporated into the revised version.

Response to Reviewer iVWF:

(i) We extend our method to segmentation tasks, where the results indicate that our method can generalize to dense visual prediction tasks.

(ii) We train ResNet50 and ViT-B from scratch to show our method's gains from GVFM adaptation.

(iii) We test multiple seeds and batch sizes to enhance the stability analysis of our method. We also introduce several orthogonality objectives to explain our choice of MINE.

(iv) We try different strategies to ease class imbalance, while finding the original setting has already been sufficient and well-performed.

Response to Reviewer Vg8X:

By calculating trainable parameters and inference cost, we find our method does not introduce significant computational overhead.

Response to Reviewer pzKj:

(i) We add the amount of learnable parameters in the appendix and find it comparable to LoRA.

(ii) We discuss the two suggested disentanglement-based medical imaging works in related work.

**Supporting Material:**

/attachment/50bfa68fe92481a0bcc9cf102d23e1a0b30536c5.pdf

---

### Meta-Review · Area_Chair_c9oQ · 2026-02-07

**Recommendation:** Accept (Poster)
**Confidence:** 5

**Metareview:**

All reviewers found the proposed method to be novel and the results promising

---

### Decision · Program_Chairs · 2026-02-13

Accept (Poster)